# Epidemiology of *Schistosoma mansoni* infection and associated risk factors among school children attending primary schools nearby rivers in Jimma town, an urban setting, Southwest Ethiopia

**Azalech Tefera[1], Tariku Belay[2], Mitiku Bajiro[2]***

**1** Laboratory Unit, Jimma Medical Center, Jimma, Ethiopia, **2** School of Medical Laboratory Sciences, Faculty of Health Sciences, Jimma University Institute of Health, Jimma, Ethiopia

\* mitikubajiro2008@yahoo.com

## Abstract

### Background

Schistosomiasis is one of the most neglected tropical parasitic disease which is common in *Ethiopia*. It is disease of rural areas for decades but now days there are reports of schistosomiasis from urban settings. Therefore, this study aimed to determine epidemiology of *Schistosoma mansoni (S. mansoni)* infection and associated determinant factors among school children attending primary schools nearby rivers in Jimma town, an urban setting, southwest Ethiopia.

### Methodology

A cross sectional study was conducted among 328 school children aged between 7–17 years in selected primary schools nearby rivers in Jimma town from March to April 2017. For the diagnosis of *S. mansoni*, a single stool sample was obtained from each child and processed using double Kato Katz thick smear for quantification of *S. mansoni* ova examined using light microscope. A questionnaire was used to collect socio demographic data and associated determinant factors for *S. mansoni* infection. Data were analyzed using SPSS version 20.0. Variables with P-value < 0.05 were significantly associated with *S. mansoni* infection.

### Results

The overall prevalence of *S. mansoni* infection was found to be 28.7%. Majority of infection intensities were categorized as light with maximum egg per gram of stool (epg) was 1728. The geometric mean of infection intensity was 102.3epg. Schools distance from river (p = 0.001), swimming habit in rivers (p = 0.001) and crossing river on bare foot (p = 0.001) were independent risk factors for *S. mansoni* infection.

**Data Availability Statement:** All relevant data are within the manuscript and its Supporting Information files

**Funding:** The study is funded by organization Jimma University institute of health research and postgraduate. So there is no specific funder with this research. The funder had no role in study design, data collection and analysis decision to publish, or preparation of the manuscript.

**Competing interests:** The authors have declared that no competing interest exist.

**Abbreviations:** AOR, Adjusted Odd Ratio; CI, Confidence Interval; COR, Crude Odd ratio; EPG, Egg per Gram; SPSS, Statistical Package for Social Science; WHO, World Health Organization.

## Conclusions

The study revealed S. *mansoni* infection is prevalent in Jimma town. The school children were at moderate risk of morbidity caused by *S. mansoni* (prevalence $\geq$ 10% and < 50% according to WHO threshold), hence a biannual mass drug administration with praziquantel is required once every two years in the study area and promote health information on prevention, control, transmission and risk factors for *S. mansoni* infection.

## Introduction

Schistosomiasis caused by *S. mansoni*, is among the most prevalent but neglected tropical parasitic diseases. It is an intravascular parasite caused by blood fluke trematode belongs to genus *Schistosoma. S. mansoni, Schistosoma japonicum* and *Schistosoma haematobium* are responsible for most cause infection in human and *Schistosoma mekeongi* and *Schistosoma intercalatum* are less prevalent species [1].

*Schistosomes* are transmitted through contact with fresh water contaminated with human excreta containing *schistosomes* eggs which hatch in fresh water and releasing free swimming miracidia; which infect aquatic snail *Biomphalaria pfeifferi*, an intermediate host for *S. mansoni* to complete its life cycle and release cercariae in to water and human can be infected during contacts with water for various domestic purposes[1].

The clinical manifestation of schistosomiasis occurs in three stages, which involves swimmers' itch, acute, and chronic schistosomiasis. Swimmers' itch is a local inflammatory reaction, visible at the site of cercarial penetration, composed of edema, dilated capillaries and a few cells, attributed to the local release of monokines in which the severity of the reaction is depending on the duration of schistosomulae stay in the dermis.

Acute schistosomiasis is due to systemic hypersensitivity reaction against the migrating larva through different system which is characterized by sudden fever, fatigue, myalgia, malaise, non-productive cough and eosinophilia, then after the immature worm complete its migration and positioned themselves to the final position they resides then followed by abdominal symptoms in most cases infected individuals recover spontaneously within few weeks but some infected individuals developed to persistent and become more serious disease with weight loss, dyspnoea, diarrhea, diffuse abdominal pain, toxaemia, hepatomegaly, splenomegaly and widespread rash.

Chronic schistosomiasis is due the eggs trapped in the tissues during the perivesical or per intestinal migration in the liver, spleen, lungs, or cerebrospinal system.

The eggs secrete proteolytic enzymes that provoke typical eosinophilic inflammatory and granulomatous reactions, which are progressively replaced by fibrotic deposits[2].The severity of the symptoms is thus related both to the intensity of infection and to individual immune responses.

Clinical signs and symptoms of schistosomiasis in children is generalized, non-specific signs and symptoms, that makes it difficult to identify disease-specific morbidity signs, thus why morbidity may progress from subtle manifestations such as anemia, to more severe, devastating and irreversible conditions such as growth stunting, impaired cognitive development, increased susceptibility to co-infection, decreased quality of life, exercise intolerance, infertility, portal hypertension, and liver failure[3].

Diagnosis of schistosomiasis is conducted based on parasitological examination of stool using Kato Katz thick smear technique which is widely used and also recommended by WHO for surveillance and monitoring program[4].

Schistosomiasis is endemic in 77 countries of tropical and sub-tropical regions of the world. It is estimated that the number of infected individuals worldwide were 237 million with >97% of infected individual live in Africa, and 120 million people already diseased; of which 20 million were severely diseased. About an estimated 779 million people are at risk of infection with 85% of the population are living in Africa. The estimated annual mortality rate is to be 150,000–280,000 and 1.7–4.5 million in disability adjusted life years lost [5–9].

Inadequate clean water supply, poor environmental sanitation and limited access to education are factors which makes intestinal helminthes infection remains a significant health problem in developing countries[10]. The high prevalence of *S. mansoni* infection in developing countries is directly associated with infested water bodies (pond, stream, river, and dam) by cercariae and contact with it during crossing with bare foot, swimming, washing of clothes and utensil, playing, fishing and irrigation activity[11].

Water temperature, absence or presence of snail intermediate host, population movement and development of water dams for irrigation and hydroelectric power are determinant factors for distribution and transmission *Schistosoma* species (*S. mansoni* and *S. haematobium*) in Ethiopia [12].

*S. mansoni* is the most prevalent species in Ethiopia and its prevalence has been reported in different regions of the country with rapid distribution in connection with water resource development and intensive population movements, with nearly 90% reported among school children [13].

Various epidemiological studies reported *S. mansoni* infection among school children in different countries with disparity in prevalence with Brazil 14.4% [14], Northwestern Tanzania, 64.3% and 84.01%[15, 16], Roryal District, Northwestern Tanzania 15.1% [17], high altitude crater lakes in Western Uganda 27.8%[18], Kismu city western Kenya 21.0% [19], Nigeria, 4.6% and 12.6% [20, 21], Ghana 19.8% [22].

Various epidemological studies in different localities of Ethiopa also reported *S. mansoni* infection with vary in prevalence. School children in Dembia district 11.2% [23], north Gonder 33.7% [24],Adarkay District 54.3% [25],Dembia plains 35.8% [26], Sanja Town 89.9% [27], Bahir Dar town 7.3% [28], Hayk Town 45% [29] from Northwest part of Ethiopia are reported to be infected with *S. mansoni*.

There were also reports from Southern part of Ethiopia with variation in prevalence from Wolayita, 58.6% and 81.3% [30, 31], Bushullo village, 73.7% [32], Lake Hawassa, 31%[33], Wendo Genet, 74.9% [34], Southeast of lake Langano. 21.2% [35], different water source users in Tigray, 5.95% [36], Waja-Timuga, District of Alamata, 73.9%[37], Suburbs of Mekelle city, Tigray, 23.9% [38], Northern part Ethiopia and Horru Guduru Wollega 67.6% [39] Western parts of Ethiopia and surrounding district of Gelgel Gibe Dam Jimma zone 2.1% [40], 24.0%; 27.6% from Jimma zone Manna district[41, 42] and 8.4% from Jimma town [43] Southwest Ethiopia.

There is evidence that *S. mansoni* infection is prevalent in Jimma Town from case reports in health facilities, presence of intermediate host in river crossing the town and water bodies near to some of the schools in which the students have contact with it at different time during schooling; however, there seems to be no document that report prevalence *S. mansoni* in Jimma town.

School children in selected primary schools in Jimma town are groups of population are more at risk of infections with *S. mansoni* since the students have frequent contacts with the river crossing the town and water bodies nearby the schools. Therefore, the objective of this study was to determine epidemiology of *S. mansoni* infection and associated determinant factors among School Children attending Primary Schools nearby Rivers in Jimma Town, an urban setting, Southwest Ethiopia.

## Materials and methods

### Study area and study population

The study was conducted between March and April, 2017 among school children from five purposely selected primary schools namely Kito, Hamile 19, Jimma primary, Seto Yido and Tesfa Tewahido that are located nearby rivers in Jimma town, Oromia regional state, Southwest Ethiopia. Jimma Town is located at altitude 1780 meters above sea level and longitude of 7˚ 40'N, 36˚ 50'E, 352 Km, to the Southwest of Addis Ababa. It is characterized by a warm climate with an average annual rainfall of 800–2,500 mm. Temperature in Jimma is within a comfortable range, with the daily mean temperature between 20˚C and 25˚C year-round and it has a tropical rainforest climate. It has features of a long annual wet season from March to October. Based on the 2007 Central Statistics Agency census report, the projected total population of the town was 134,040. Based on Jimma town health office record, there are two governmental hospitals, four health centers, and seventeen urban health extensions are found in the town. According to information obtained from Jimma town educational office, there are 40 primary schools in the town, 22 governmental and 18 non-governmental primary schools. There are rivers in Jimma town which are considered as potentially risk factors for *S. mansoni* infection as the students have frequent contact with it for domestic purpose. Among these Awetu River which crosses the town dividing the town in to two half junction at different sites in which the community as well the students have contact with the river and Jimma primary school is close to it. Seto and Kaba rivers are near to Seto yido and Tesfa Tewahido primary schools. Kito River is near to Hamile 19 and Kito primary schools. School children aged from 7 to 17 attending school in selected primary schools nearby rivers in Jimma town were involved in the study.

### Study design and sample processing

School based cross-sectional study was employed among five primary school children nearby rivers in Jimma town. The schools were purposely selected based on their proximity to rivers and water bodies. The school children were supplied with dry and clean-labeled plastic container cup and tissue paper and advised to bring proper stool samples with the container. All the specimens were checked for their label and quantity. For the benefit of the students' wet mount microscopic examination was done at schools to examine motile trophozoite stage of protozoan parasitic infection. Then the stool samples were transported and processed for microscopic examination in the laboratory of Medical Parasitology, School of Medical Laboratory Sciences in Jimma University, for screening of ova of *S. mansoni*.

Kato-Katz parasitological technique was applied according to the WHO recommendation for the quantification of S. *mansoni* eggs in stool samples. A double Kato Katz slides were prepared from single stool sample collected from each child. The Kato -Katz preparations were examined after 24 hours for quantification of ova of *S. mansoni* and other soil transmitted helminthes except hookworm; which was examined within 30 minutes to an hour.

### Clinical examination

Experienced Nurses and Public health officers assessed the presence or absence of signs and symptoms of *S. mansoni* infection like, abdominal pain, fever, palpable splenomegaly and hepatomegaly, and fatigue.

### Sample size and sampling technique

Sample size was determined using single population proportions using P = 26.3% from the research done on communities living in Jimma town near by three rivers in the town [44] by

using the below formula:

$$n = \frac{\left(z\frac{z}{2}\right)^2 p(1-p)}{d^2},$$

Where, n is the sample size, Z is statistics for level of confidence, P is expected prevalence or proportion, and d is measures the Precision of the estimate and the calculated sample size was 298 and considering 10% non-response rate the final sample size was 328.

Systematic random sampling technique was used to select school children who participated in the study based on proportional allocation of students from the five primary schools using class roaster/attendance that contains complete list of students from the schools before commencing data collection.

Following allocations, students in each class were ordered alphabetically and the first primary student was selected in random starting point and the next students were included in a fixed periodic interval. When the selected student was absent, the student after the indicated one was sampled for replacement. In each school, we stratified students according to three age groups (7–10 years, 11–14 years and 15–17 years). Below is the figure indicating how the students were proportionally allocated and selected from the five primary schools nearby rivers in Jimma town (Fig 1).

## Data processing and statistical analysis

Data were coded, entered and cleaned by using EPIIFO version 4.0.2.101. The processing and analysis of the data were carried out using SPSS version 20.0 software. The prevalence of *S. mansoni* was presented in percent and infection intensity of S. *mansoni* was categorized as low, moderate and heavy based on WHO guide line and mean egg count was reported by geometric mean respectively. The association between *S. mansoni* infection and associated risk factors were statistically tested using binary logistic regression and the level of significance was set at P-value <0.05). The magnitude of association was measured using adjusted odds ratio (AOR), at 95% CI.

## Data quality control

Refreshment training was given for data collectors and laboratory technologist/technician about Kato-Katz thick smears by experienced laboratory technologist in the field. During data

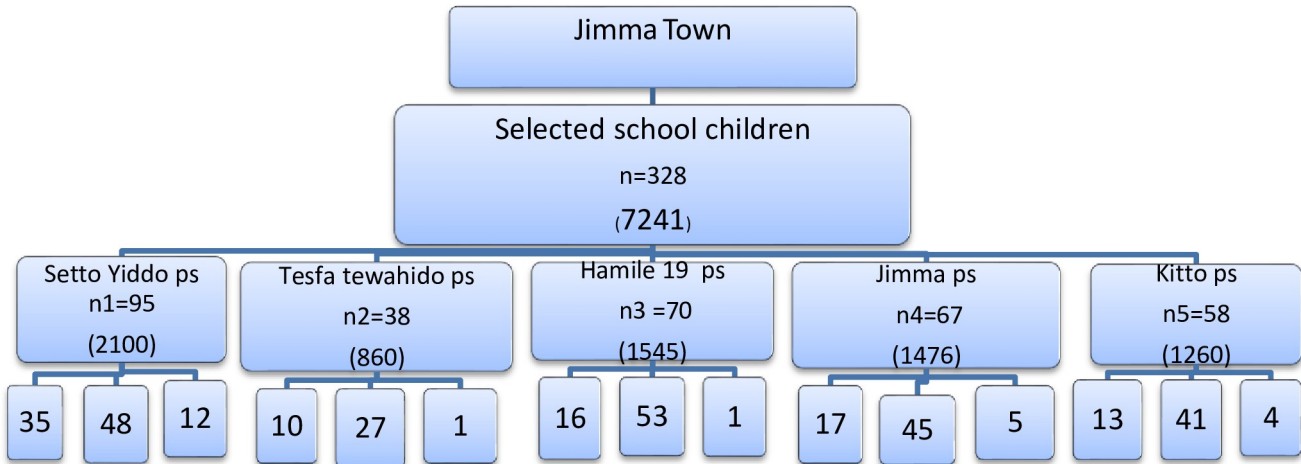

**Fig 1. How students were proportionally allocated and selected from the five primary schools nearby rivers in Jimma town, Southwest, Ethiopia, 2017.**

processing, the quality of data was assured by coding and double entry. From both positive and negative Kato-Katz smears 10% were randomly selected and re-read by two independent medical laboratory experts who are blind to the primary result. Moreover, fresh working solution of malachite-green was used routinely to maintain the quality of the smear.

### Ethical approval and consent to participate

The study was approved by Ethical review committee of Jimma University Institute of Health research and postgraduate study with Ref No. **IHRPGC/309/2017**. The purpose of the study was communicated with Jimma Town Health Officers and then with schools principals. Informed consent was obtained from the parents/guardians of the students. Students infected with *S. mansoni* were treated with 40 mg/kg Praziquantel and those infected with soil transmitted helminthes and parasitic protozoan were treated with 400 mg albendazole and Metronidazole 250mg/dose orally according to National treatment guide line of Federal Ministry of Health of Ethiopia by trained and experienced Nurses and Health Officer.

## Results

### Socio-demographic characteristics of study participants

A total of 328 school children, of which 59.8% (196) males and 40.2% (132) females from five primary schools nearby rivers in Jimma town were involved in this study. Students from the age group of 7–17 years with mean age of 11.63 were involved in the study. The largest proportion 214(65.2%) of study participants were selected from the age group of 11-14(Table 1).

## Prevalence of *S. mansoni*

Stool samples were collected from 328 school children and examined for presence of *S. mansoni* and other intestinal parasites. From 328 school children involved in this study 218(66.5%) children were found to be infected with at least one parasite species. In addition to *S. mansoni*, *T. trichiura 66(20.12%)*, *A. lumbricoides 16(4.9%)*, *H. nana 10(3.04%)*, hookworm 1(0.3%), Teania species 1(0.3%), *G. lambelia* 15(4.6) and *E. histolytica/dispar10(10.04%)* were also detected in the stool samples examined.

**Table 1. Socio-demographic characteristics of school children in selected primary schools nearby rivers in Jimma town, an urban setting, Southwest Ethiopia, 2017.**

| Characteristics | |
|---|---|
| Gender | n(%) |
| Male | 196(59.8) |
| Female | 132(40.2) |
| Age groups (years) | |
| 7–10 | 91(27.7) |
| 11–14 | 214(65.2) |
| 15–17 | 23(7.1) |
| Schools | |
| Seto Yiddo | 95(29.0) |
| Jimma primary school | 67(20.4) |
| Hamile 19 | 70(21.3) |
| Kitto | 58(17.7) |
| Tesfa Tewahido | 38(11.6) |

The overall prevalence of *S. mansoni* in the selected primary schools nearby rivers in Jimma twon was found to 28.7% (94/328). The prevalence was 39.3% (77/196) and 12.9% (17/132) between male and female school children respectively. School children age groups between 15-17(39.1%) was more infected. The prevalence vary from 12.1% to 38.9% among the schools with the highest prevalence 38.9%(37/95) in Seto Yiddo school (Table 2).

## Infection intensity of *S. mansoni*

Among 328 school children examined for presence of *S. mansoni* infection 94 school children excreted ova *S. mansoni* and the ova was quantified based on WHO guideline by counting epg stool. From these school children excreting ova of *S. mansoni* 50(53.2%), 33(35.1%), 11(11.7%) of them were categorized as light, moderate and heavy infection intensity. The majority of infection intensity was categorized as light infection intensity with maximum of 1728 epg. The geometric mean of egg count was 102.3 epg (Fig 2).

## Clinical manifestation of *S. mansoni*

School children participated in this study undergone physical examination and diagnosed for sign and symptoms related with schistosomaisis. The most frequent clinical investigation from school children was abdominal pain 162 (49.4%) (Fig 3).

61.0% (200/328) had one or more clinical sign and symptoms of *S. mansoni* infection. The remaining 128(39.0%) had no clinical manifestations. 24.5% (49/200) of the school children with clinical presentation were positive for *S. mansoni* and 75.5% (151/200) them found to be negative for *S. mansoni* infection respectively (Fig 4).

Of 39.0% (128/328) of school children with no clinical manifestations 35.2% (45/128) were positive for *S. mansoni* infection and 64.8% (83/128) were found to be negative for *S. mansoni* infection (Fig 5).

## Factors associated with *S. mansoni* infection

Bivariate factor analysis was carried out for association of *S. mansoni* infection and factors among school children such as age in year, gender, distance of school from river, swimming

**Table 2. Prevalence of *S. mansoni* among selected primary school children nearby rivers in Jimma town, an urban setting, Southwest Ethiopia 2017.**

| Variables | *S. mansoni* infection status | | Total (%) |
|---|---|---|---|
| | Number of positive (%) | Number of negative (%) | |
| Gender | | | |
| Male | 77(39.3) | 119(60.7) | 196(59.8) |
| Female | 17(12.9) | 115(87.1) | 132(40.2) |
| Age (years) | | | |
| 7–10 | 21(23.1) | 70(76.9) | 91(27.7) |
| 11–14 | 64(29.9) | 150(70.1) | 214(65.2) |
| 15–17 | 9(39.1) | 14(60.9) | 23(7.1) |
| Schools | | | |
| Seto Yiddo | 37(38.9) | 58(61.1) | 95(29.0) |
| Jimma primary school | 16 (23.9) | 51 (76.1) | 67(20.4) |
| Hamile 19 | 26(37.1) | 44 (62.9) | 70(21.3) |
| Kitto | 7(12.1) | 51(87.9) | 58(17.7) |
| Tesfa Tewahido | 8(21.1) | 30 (78.9) | 38(11.6) |

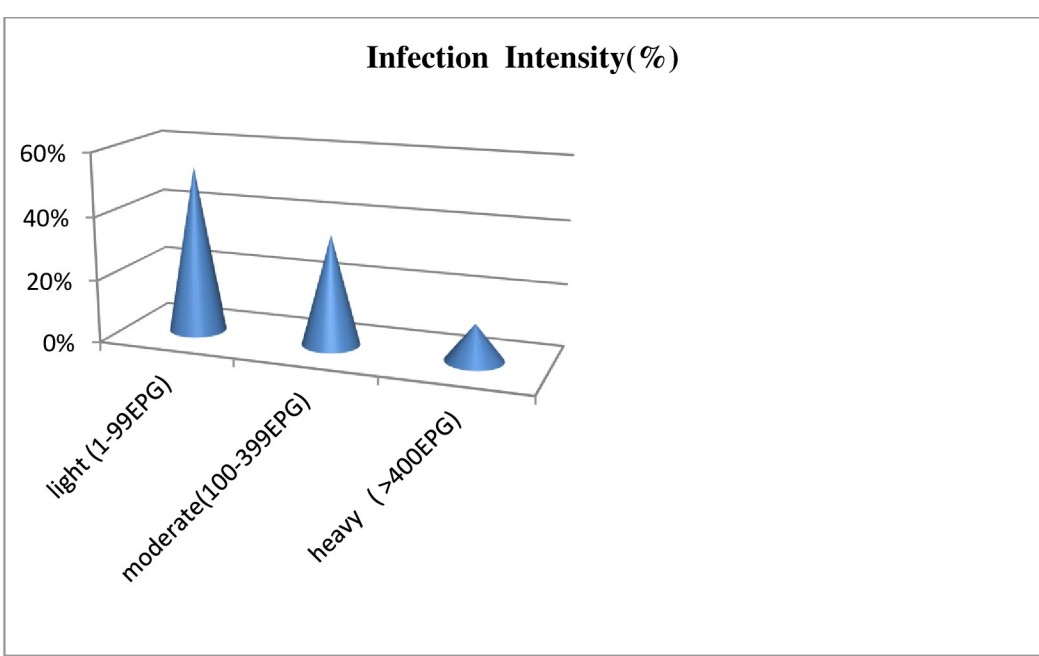

**Fig 2. Infection intensity of *S. mansoni* among purposively selected primary school children in Jimma town, Southwest Ethiopia, 2017.**

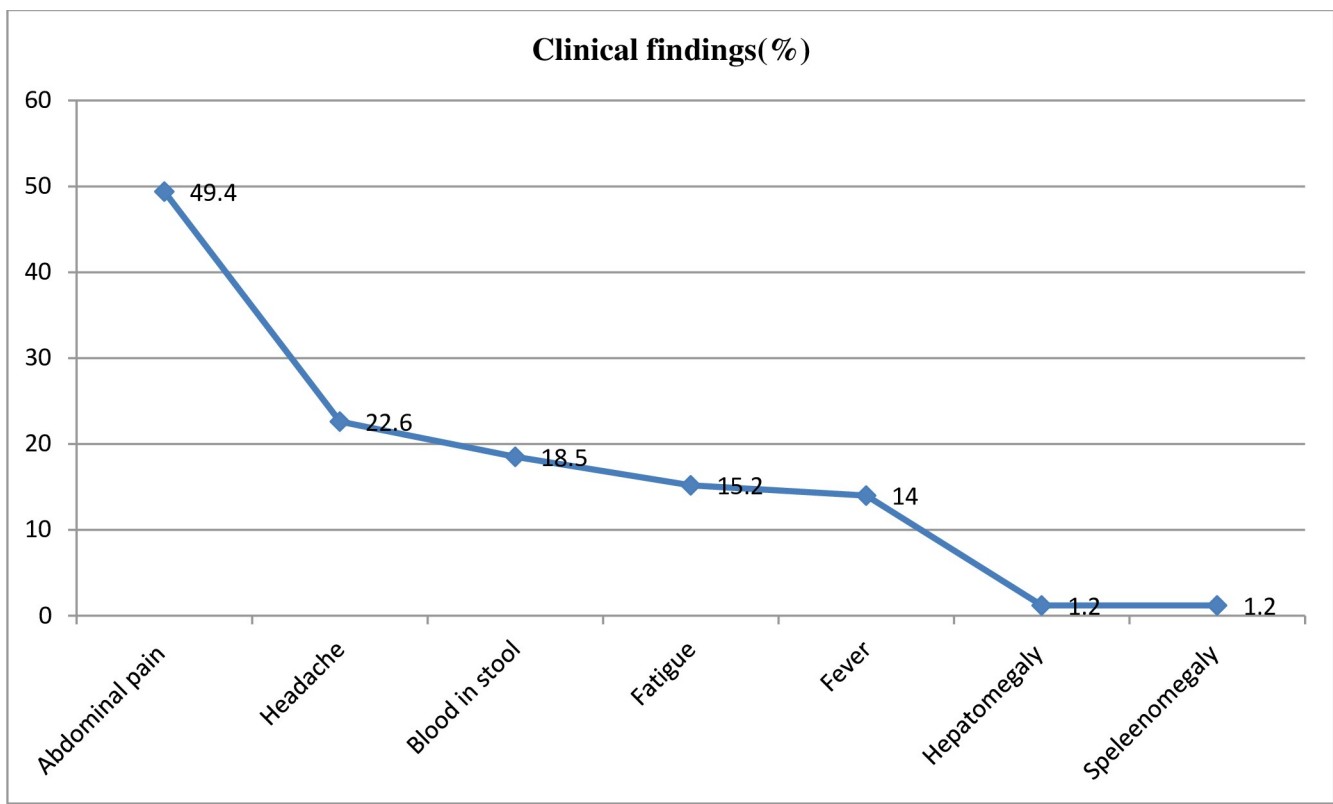

**Fig 3. *S. mansoni* positive with clinical manifestation among primary school children attending school nearby rivers in Jimma town, an urban, Southwest Ethiopia, 2017.**

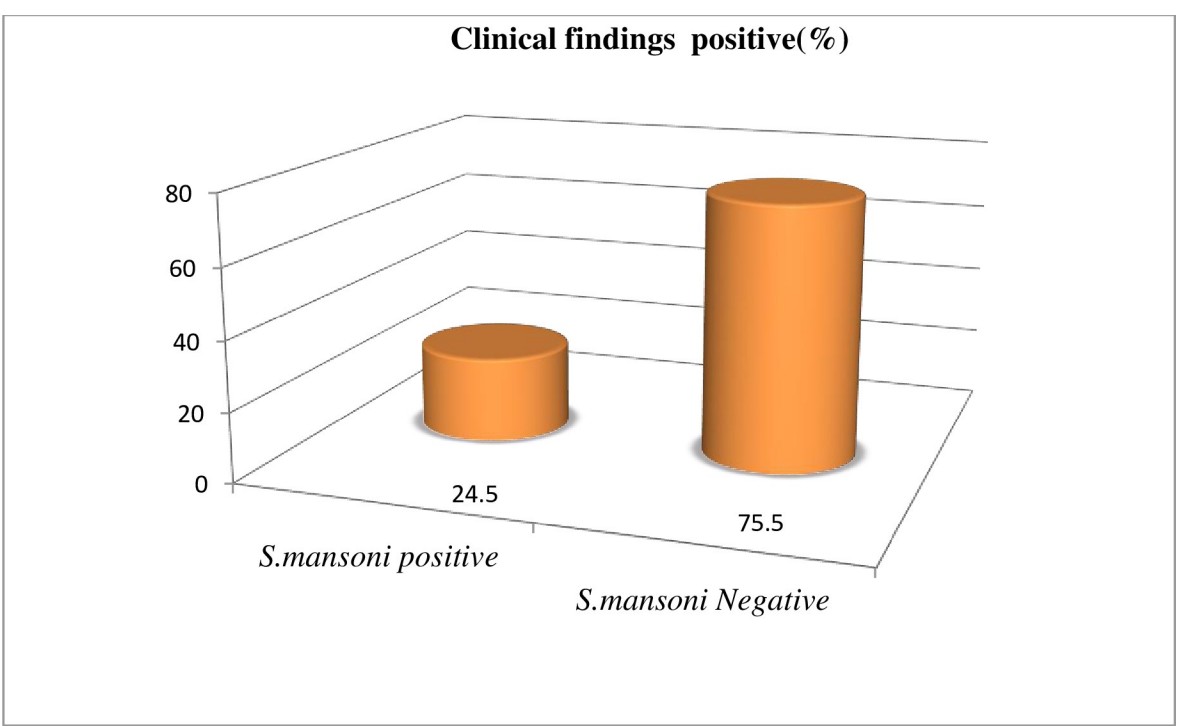

**Fig 4. Prevalence of *S. mansoni* infection with clinical findings among school children attending primary schools nearby rivers in Jimma town, an urban setting, Southwest Ethiopia, 2017.**

habit, swimming frequency, river contact while crossing, washing clothes in the river, habit of bathing in the river and variables with P-value less than 0.025 were used as cut off point for considering as candidates for multiple logistic regression to identify independent risk factor for *S. mansoni* infection(Table 3).

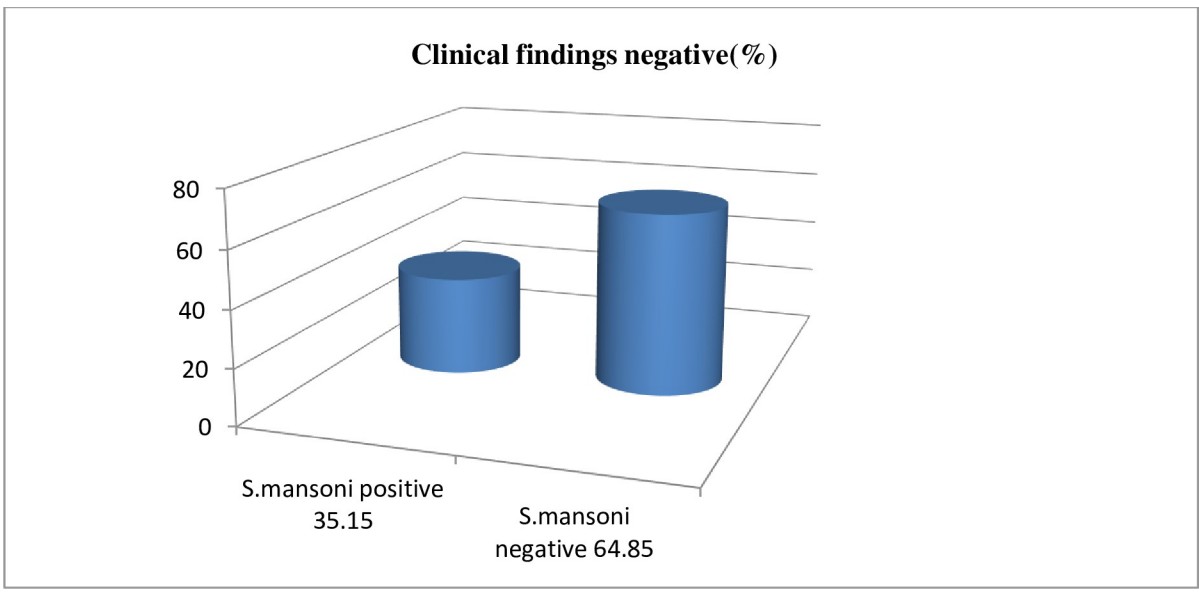

**Fig 5. Prevalence of *S. mansoni* infection with negative clinical findings among school children attending primary school nearby rivers in Jimma town, an urban setting, Southwest Ethiopia, 2017.**

**Table 3. Bivariate analysis of factors associated with *S. mansoni* infection among school children attending primary schools nearby rivers in Jimma town, an urban setting, Southwest Ethiopia, 2017.**

| Risk factors | | No. of infected | No. of not infected | Bivariate | | | |
|---|---|---|---|---|---|---|---|
| | | | | 95% CI for COR | | | |
| | | | | Total | Odds ratio | 95% CI | P-value |
| Age in year | 7–10 | 21 | 70 | 91 | 1:00 | | |
| | 11–14 | 64 | 150 | 214 | 1.422 | 0.805–2.512 | 0.225 |
| | 15–17 | 9 | 14 | 23 | 2.143 | 0.813–5.647 | 0.123 |
| Gender | Male | 77 | 119 | 196 | 4.377 | 2.440–7.852 | 0.001 |
| | Female | 17 | 115 | 132 | 1:00 | | |
| Distance of School from river | School 1 | 37 | 58 | 95 | 4.648 | 1.906–11.331 | .001 |
| | 2 | 8 | 30 | 38 | 1.943 | 0.640–5.896 | 0.241 |
| | 3 | 26 | 44 | 70 | 4.305 | 1.704–10.878 | 0.002 |
| | 4 | 16 | 51 | 67 | 2.286 | 0.867–6.025 | .095 |
| | 5 | 7 | 51 | 58 | | 1.00 | |
| Swimming habit | Yes | 80 | 124 | 204 | 5.069 | 2.718–9.453 | 0.001 |
| | No | 14 | 110 | 124 | | 1.00 | |
| Swimming frequency | Daily | 7 | 26 | 33 | 0.365 | 0.762–5.664 | 0.26 |
| | Weekly | 73 | 98 | 171 | | 1.00 | |
| Crossing river on bare foot | Yes | 66 | 85 | 151 | 4.132 | 2.466–6.923 | 0.001 |
| | No | 28 | 149 | 177 | | 1.00 | |
| Washing clothes in the river | Yes | 40 | 83 | 123 | 1.348 | 0.827–2.197 | 0.232 |
| | No | 54 | 151 | 205 | | 1.00 | |
| Habit of bath in the river | Yes | 27 | 55 | 82 | 1.312 | 0.765–2.249 | 0.324 |
| | No | 67 | 179 | 246 | | 1.00 | |
| Total | | 94 | 234 | 328 | | | |

Among variables used as candidates for multiple logistic regression gender (AOR: 3.31, 95%CI, 1.69–6.46, P = 0.001), crossing rivers on bare foot(AOR: 3.17, 95%CI, 1.73–5.80, P = 0.001), swimming or playing nearby river(AOR = 4.68, 95%CI, 2.37–9.23, P = 0.01), distance of school from river, were significantly associated and independent predictors for *S. mansoni* infection (p<0.05) (Table 4).

**Table 4. Multivariate factor analysis associated with *S. mansoni* infection among school children attending primary school nearby rivers in Jimma town, an urban setting, Southwest Ethiopia, 2017.**

| Variables | | No. of infected | No. of not infected | Total | Multivariate | |
|---|---|---|---|---|---|---|
| | | | | | AOR 95%CI | P-value |
| Gender | Male | 77 | 119 | 196 | 3.312(1.691–6.486) | 0.001 |
| | Female | 17 | 115 | 132 | 1:00 | |
| Distance of school from river | School 1 | 37 | 48 | 95 | 5.111(1.985–13.163) | 0.001 |
| | 2 | 8 | 30 | 38 | 5.389(1.564–18.563) | 0.008 |
| | 3 | 26 | 44 | 70 | 7.254(2.668–19.721) | 0.001 |
| | 4 | 16 | 51 | 67 | 3.433(1.233–9.559) | 0.02 |
| | 5 | 7 | 51 | 58 | 1.00 | |
| Swimming habit | Yes | 80 | 124 | 204 | 4.679(2.372–9.231) | 0.001 |
| | No | 14 | 110 | 124 | 1.00 | |
| Crossing rivers on bare foot | Yes | 66 | 85 | 151 | 3.166(1.729–5.800) | 0.001 |
| | No | 28 | 149 | 177 | 1.00 | |

## Discussion

In the present study, the prevalence of S. *mansoni* infection among five selected primary school children's nearby rivers in Jimma town was 28.7% with the prevalence between male and female was 39.3% and 12.95% respectively. Majority of infection intensity were classified as light with maximum of 1728 epg. Gender, distance of school from river, river water contact while crossing, swimming or playing nearby river were independent predictors for *S. mansoni* infection.

The prevalence in the present study was almost comparable with the study from Jimma town 26.3% [44], 25.3% [45], 24% & 27.6% from Manna district Jimma zone[41, 42], 27.6% from Saja town Amhara region[46] 33.7% in Gonder[24],31% from lake Hawassa[33], 23.9% from suburbs of Mekelle city, Tigray[38], 27.8%from high altitude crater lakes in Western Uganda[18].

The prevalence rate in the present study was higher than the one reported from Jimma zone, nearby Gilgel Gibe dam, 10.61% and 2.1%[40, 47],8.4% Jimma town [43], 21.2% from Southeast of lake Langano[35], 7.3% from Bahir Dar town [28], 20.6% from Gorgora town Northwest Ethiopia [48], 20.2% in Raya, Alamata[49], 5.95% in Tigray from different water sources [36], 0.8% in Amibera[50]. Similarly from different part of Latin America and Africa with lower prevalence were recorded from Brazil 14.4% [14], Nigeria, 4.6% and 12.6%, Ghana 19.8% [22], 18.9% in Sierra Leone [51], and 1.1%, 12.5% in two endemic area of Niger River basin[52], 1.5% from Bamako, Mali[53],15.1% of S. *mansoni* from Roryal District, Northwestern Tanzania[17],21.0% of S. *mansoni* from Kismu city western Kenya [19], 12.1% from Agaie, Niger State, Nigeria[20], 20.3% Birnin-Gwari local government area Kaduna state, Nigeria [54].The difference might be due to the study period, the laboratory diagnosis employed, sample size used, environmental factors which favor distribution of intermediate host, proximity water bodies and longtime endemicity in the area.

On the other hand, the prevalence of *S. mansoni* infection in the present study lower than reports Sengerema District, North Western Tanzania 64.3% and 84.01% from Northwestern Tanzania [15, 16], 69% from Nyanza province Kenya [3]. Similarly from different parts of Ethiopia 45% from Hayk [29],58.6% and 81.3% from Wolita [30, 31],73.7% from Bushullo village [32]74.9% from Wondo Genet[34],65.5%, 55% in Wondo Genet [55], 89.9% from Saja town Northwest Ethiopia [27], 67.6% from Wollega[39].This variation might be probably because of water source development in those areas, water contact behavior of children, differences in climatic conditions of the area, distance of schools from water sources, Study periods and duration of endemicity in that area.

Majority of infection intensity in the study area was categorized as low infection intensity which is similar with study reported from Wollo, Hayke [29], Wondo Genet [55], Gorgora town Northwest Ethiopia[48], suburbs of Mekelle city, Northern Ethiopia [38],Tumuga and Waja, North Ethiopia [56], Jimma town [44], Manna district Jimma Zone [41], Nyanza Province Kenya [3], School- age children in rural communities from Northern Ghana [22], North western Tanzania [15] and different in which the infection intensity was moderate infection intensity from Kamissie, Wollo [55], Wollega [39], Wondo Genet [34], Saja area, Amhara region [46], Sierra Leon [57] and Yemen [58]. This variation might be due to difference in exposure to water bodies, methods used diagnosis and may be skill of professionals involved in the examination.

Male students were more infected than female students in this study [AOR = 3.312,1.691–6.486, 95%CI, P = 0.001], which is in line with study reported from Jimma town [43, 44], Wollo, Hayk area [29], Amibera district Northern Ethiopia [50] and North Ghana [22], Jos, Nigeria [21].This difference might be due to repeated water-contact activity by male children

than female when they were playing in the field and also may be due to the fact that females are mostly restricted to home with household responsibilities with their families.

Gender, distance of schools from river, crossing river/water on bare foot, swimming habits in river/ponds were independent predictors for *S. mansoni* infection which is in line with the variables crossing rivers on bare foot reported from the studies, Manna district Jimma Zone [42], Gorgora town Northwest Ethiopia [48], Saja town, Northwest Ethiopia [27], Israel [59], Swimming habits similar with study from Saja town [27], Raya Alamata District Northern Ethiopia [49], Amiebera Distrct northern Ethiopia [50], Saja area, Amahara region [46], and distance of schools from rivers similar with findings from Nigeria [54] and in Yemen [58].

## Conclusion

The school children in five selected primary schools in Jimma town were at moderate risk of the morbidity caused by *S. mansoni*, hence a biannual mass drug administration with praziquantel is requiredonce every two years in this area and promote health information on prevention, control, transmission and risk factors for *S. mansoni* infectionin the area and improving environmental sanitation and personal hygiene via construction of latrines in schools. Constructing local bridges on Awetu and Seto rivers which have role in reducing chance of contacts with infested water bodies while crossing the rivers may minimize the risk of acquiring *S. mansoni* infection.

## Supporting information

**S1 Questionnaire. Survey questionnaire used for assessing participants behaviors in English and local language.**
(DOCX)

**S1 File. Ethical approval letter to from Jimma university institute of health research and post graduate office to conduct the research.**
(DOCX)

**S2 File. Written consent form to participate in the study English version.**
(DOCX)

**S3 File. Written consent form to participate in the study in local language Afan oromo version.**
(DOCX)

**S1 Data. Laboratory procedure for Kato Katz.**
(DOCX)

## Acknowledgments

We are grateful for School teachers, study participants, and their parents. Additionally, we would like to thank the staffs of NTDs Laboratory in Jimma University for providing Kato-Katz kits and reagents.

## Author Contributions

**Conceptualization:** Azalech Tefera, Tariku Belay, Mitiku Bajiro.

**Data curation:** Azalech Tefera, Tariku Belay.

**Formal analysis:** Azalech Tefera, Tariku Belay, Mitiku Bajiro.

**Investigation:** Azalech Tefera, Mitiku Bajiro.

**Methodology:** Azalech Tefera, Mitiku Bajiro.

**Supervision:** Azalech Tefera, Mitiku Bajiro.

**Writing – original draft:** Azalech Tefera, Mitiku Bajiro.

**Writing – review & editing:** Azalech Tefera, Tariku Belay, Mitiku Bajiro.

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
