## [Decision Letter · Decision Letter 0]

29 Aug 2019

PONE-D-19-19521

Epidemiology of Schistosoma mansoni infection and associated determinant factors among School Children attending Primary Schools nearby Rivers in Jimma Town, an urban setting, Southwest Ethiopia

PLOS ONE

Dear Weltagi

Thank you for submitting your manuscript to PLOS ONE. After careful consideration, we feel that it has merit but does not fully meet PLOS ONE’s publication criteria as it currently stands. Therefore, we invite you to submit a revised version of the manuscript that addresses the points raised during the review process.

We would appreciate receiving your revised manuscript by  October 15, 2019 To enhance the reproducibility of your results, we recommend that if applicable you deposit your laboratory protocols in protocols.io, where a protocol can be assigned its own identifier (DOI) such that it can be cited independently in the future. For instructions see: http://journals.plos.org/plosone/s/submission-guidelines#loc-laboratory-protocols

We look forward to receiving your revised manuscript.

Kind regards,

Salah A Sheweita

Academic Editor

PLOS ONE

Journal Requirements:

2. We note that you have reported significance probabilities of 0 in places. Since p=0 is not strictly possible, please correct this to a more appropriate limit, eg 'p<0.0001'.

3. Please include copies of the survey questions or questionnaires used in the study to assess participant behaviours (eg. swimming, river contact, washing), in both the original language and English, as Supporting Information, or include a citation if they have been published previously.

4. We noticed you have some minor occurrence(s) of overlapping text with the following previous publication(s), which needs to be addressed:

https://www.longdom.org/open-access/schistosoma-mansoni-infection-prevalence-and-associated-determinantfactors-among-school-children-in-mana-district-jimma-zone-oromi-2155-9597-1000329.pdf

https://dx.doi.org/10.4269%2Fajtmh.2012.12-0397

https://doi.org/10.1016/S0140-6736(06)69440-3

http://dx.doi.org/10.1155/2014/792536

https://path.upmc.edu/cases/case622/dx.html

In your revision ensure you cite all your sources (including your own works), and quote or rephrase any duplicated text outside the Methods section. Further consideration is dependent on these concerns being addressed.

5.  We suggest you thoroughly copyedit your manuscript for language usage, spelling, and grammar. If you do not know anyone who can help you do this, you may wish to consider employing a professional scientific editing service.  

Reviewers' comments:

Reviewer's Responses to Questions

**Comments to the Author**

1. Is the manuscript technically sound, and do the data support the conclusions?

Reviewer #1: Partly

Reviewer #2: Yes

Reviewer #3: No

2. Has the statistical analysis been performed appropriately and rigorously? 

Reviewer #1: No

Reviewer #2: Yes

Reviewer #3: Yes

3. Have the authors made all data underlying the findings in their manuscript fully available?

Reviewer #1: Yes

Reviewer #2: Yes

Reviewer #3: No

4. Is the manuscript presented in an intelligible fashion and written in standard English?

Reviewer #1: No

Reviewer #2: Yes

Reviewer #3: Yes

5. Review Comments to the Author

Reviewer #1: This manuscript reports the epidemiological characteristics of schistosomiasis among schoolchildren nearby rivers in Jimma Town, Southwest Ethiopia. The study is too simple and the number of targeting people is 328 schoolchildren (relatively small). Several concerns are as follows:

1. The survey was done only one time targeting 328 schoolchildre using a single stool sample from each child; questionnaire study was also done. However, the results show no unique/novel findings compared with previous ones done in the similar areas (references no. 40, 41, 42, 43).

2. The descriptions are not so well done. For example, in introduction part, the textbook level information on schistosomiasis is described unnecessarily as long paragraphs.

3. Reference citations are too many; the number can be reduced into less than a half.

4. English should be improved. A native speaker should consult on English of this manuscript.

Reviewer #2: In this study, Tefera and colleagues investigated the prevalence of S. mansoni infection and the associated risk factors among schoolchildren in Jimma Town, Ethiopia. This manuscript is a contribution to the literature, but there are some issues, that if address, would improve it:

Points:

1) Authors mentioned that only a single stool sample was collected from each child. This may lead to inaccurate results as according to CDC guideline three samples should be collected on different days to increase the sensitivity of stool diagnosis. Negative findings reported in this survey may not reflect the true prevalence of the disease.

2) Authors also reported that several other helminth and protozoa eggs were detected in the stool, however, these data were not included in the manuscript. This data should be included to show the co-infection status among these schoolchildren, which is equally important.

3) Discussion – Authors should discuss in more detail on their findings rather than just comparing the prevalence in %.

4) There are some grammatical errors which require English editing.

e.g. schistosomes – not italic

sex – change to gender

hepatomegaly, splenomegaly

Reviewer #3: The study reports Schistosoma mansoni infection among school children and associated risk factors in an urban setting of Ethiopia. The manuscript adds useful information on distribution of schistosomiasis in Ethiopia where new foci of schistosomiasis transmission is being reported. However, the manuscript has some major issues in particular the second aspect aiming to address “associated determinants factors”.

Frist, I would suggest not using ‘determinant’ in the title as the study was not really designed to address/identify ‘determinants’ of S. mansoni infection, more appropriately, ‘likely risk factors’, or ‘risk factors’ at best.

Second, since the analysis (addressing risk factors) was performed at the whole town scale and study subjects recruited from a small subset )five primary schools) of schools the in the town. It would be helpful/important to know if the ‘five school’ are presentative of the schools in the town in the context of schistosome infection. Therefore more information on the five schools (e.g. geographical location and environment factors of their corresponding surroundings) are important to readers. Would it be possible to add a map showing the distribution of all schools and study school, rivers etc.? That would be very helpful.

I am concerned about the clinical manifestation aspect of survey. As many surveyed variables (e.g. headache etc.) are not schisto specific and the results didn’t seem to produce any meaning information. It doesn’t seem to me this aspect adds any value to the paper.

Many descriptions seem to be redundant and close attention should be paid to the language, e.g. increasing clarity and avoid redundancy)

6. PLOS authors have the option to publish the peer review history of their article (what does this mean?). If published, this will include your full peer review and any attached files.

Reviewer #1: No

Reviewer #2: No

Reviewer #3: No

---

## [Author Response · Author response to Decision Letter 0]

28 Oct 2019

Academic editor and reviewers comments

Academic editor Response 

 � Significance of probabilities P<0.000 � The comment is accepted and corrected to P<0.0001 in the Table 3 between line 304 and 305 

 � Survey questions or questionnaires used in the study to assess participant behaviors (eg. swimming, river contact, washing) � The comment is accepted and survey questionnaire both in local language and English version attached as supporting information 

 � overlapping some text in previous publications � With regard to overlapping of some text in one of the publication I am the author and the study area is different from the current even the objective is different meaning the current one is to assess the distribution of S. mansoni among school children near to water bodies in an urban setting where as the previous one was done in the rural areas. The risk factors where ever it is the same. With others publications mentioned I used the figures from the articles for review and they were done somewhere else. Overall the current research focus on school children near to water bodies in the town and the assess the effect of water on S. mansoni we have published one articles before in the town which reported prevalence of S. mansoni in which the risk factors weren’t assessed and second there is variation among the schools in prevalence and now we have focused on school children near to water bodies to see the effect of water bodies and association with S. mansoni 

Reviewer #1 Response 

Epidemiological study issue � Yes we have done epidemiological study in the town and we have reported prevalence data and associated risk factors for S. mansoni infection. With regard to the survey it isn’t simple now days mass drug administration was implemented in different areas of the country and our study was done in urban settings and in this areas in most cases mass drug administration wasn’t implemented at the time of data collection and it gives hint for Jimma town health office to implement mass drug administration

Sample size issue � With regard to sample size we can’t say the sample size is too small as we determined sample size based sample size calculation formula. The other issue we can’t screen all school children because of resource and time and human power also. Thus why we have used some previously published article and used P value 26.3% from the research done in communities living in Jimma town near by three rivers in the town. So we have followed recommended procedure for determining sample size for doing research. 

 � Results compared with previous study done in the area �Reference 40 the research was done in the rural area of the zone on intestinal helminthiasis surrounding districts hydro electric dam in which the prevalence was not compared with the current study which was 2.1% vs 28.7%. Even the school children considered as low risk versus moderate risk. 

With regards to no unique findings with references mentioned � Reference 41 the study objective is different from the current one and there was repeated infection from registration book in the health facility and we suspect drug resistance and we did drug efficacy meanwhile we determine prevalence but the study area was rural vs urban in which the setting different. 

The same is true for reference 42 except we have assessed the risk factors in the same area with 41 with slight difference with prevalence.

Reference 43 the study was in the same area in the town and only prevalence data was reported and the risk factors weren’t assessed. In the previous study we have tried to include all the school children in the town and we have reported there great variation in prevalence of S. mansoni among the schools and those schools near to water bodies were more infected and thus why we took this five school and we have tried to report both risk factors and prevalence and the prevalence was different 8.4% in previous and 28.7% in the current. Even had it been more samples were screened the prevalence was may be more than this 

With regards to single stool samples � Yes we have collected single stool sample from school children but we have prepared double kato katz from single stool samples based on WHO guideline recommendation for screening of intestinal parasitosis. We have used double Kato katz to decrease the chance of missing the ova and increasing the rate of prevalence as per recommendation of WHO guideline.

 � Long paragraph from line 85-97 in the introduction part � The comment is accepted and I have tried to shorten the lines.

 � Reference too many � There is no restriction of reference and others in the guideline and all the references I have used are relevant with our manuscript for discussion and even for literature review. 

 � English should be improved � The comment is accepted and I have tried to improved it 

Reviewer #2 Response

 � Tefera and his colleagues investigation on S. mansoni infection and its associated factors among school children in Jimma town � The comment is accepted but I couldn’t find the article thus why I didn’t include either in review literature or discussion. 

CDC guideline three samples examined for three different days � Examination of stool for three days on every other day is recommended for diagnostic purpose not for screening of large sample size. We have used WHO guideline published on 2014 which recommended taking single stool sample and examine using single thick Kato Katz for Schistosoma mansoni and soil transmitted helminthiasis and we have used double Kato Katz to increase sensitivity. It is most difficult to take three samples from each student at day interval of three days due to resources and time too. Therefore the comment raised by reviewer is used for diagnostic purpose not for research case. 

 � About other helminthes and protozoa detected in the stool � The comment is accepted and the data with specific Species of parasites were included in the manuscript line 242 to 247 

 � About discussion � In the first paragraph of discussion I have tried to discuss our findings, then I have discussed with other findings done before either in the country or abroad that is what we have to do under discussion and up to my understanding the discussion is well discussed with similar research done on school children.

 � About schistosomes italics � The comment accepted and it is italicized and corrected 

Sex � The comment accepted and corrected then changed to gender 

Hepatomegally and splenomegally � The comment accepted and corrected as hepatomegaly and splenomegaly line 187 and 188

Reviewer#3 Response

About associated determinants factors � The comment is accepted and changed to risk factors 

School representativeness � Yes these schools were selected based on proximity to water bodies with near the schools and river crossing the town and schools were purposely selected but the students were by systematic random sampling technique which has no bias on students selection with proportional allocation was made to each school. Thus the result derived from the schools can be representative 

The school environment is close to the water bodies in which the students have contact with these water bodies during their vacation and playing in the field in the compound and the snail serving as intermediate host is available in water bodies and identified as they are infected. Thus in the surrounding environment there are risk factors for infection for S. mansoni. In the previous study we have reported prevalence S. mansoni in the town but we didn’t assessed the risk factors and now we included assessing risk factors focusing on schools closed to water bodies. 

Clinical manifestation specific with headache � The comment accepted and the headache deleted between line 186 and 188

---

## [Decision Letter · Decision Letter 1]

7 Jan 2020

Epidemiology of Schistosoma mansoni infection and associated risk factors among School Children attending Primary Schools nearby Rivers in Jimma Town, an urban setting, Southwest Ethiopia

PONE-D-19-19521R1

Dear Dr. Weltagi,

We are pleased to inform you that your manuscript has been judged scientifically suitable for publication and will be formally accepted for publication once it complies with all outstanding technical requirements.

With kind regards,

David Joseph Diemert, M.D.

Academic Editor

PLOS ONE

Additional Editor Comments (optional):

Reviewers' comments:

Reviewer's Responses to Questions

**Comments to the Author**

1. If the authors have adequately addressed your comments raised in a previous round of review and you feel that this manuscript is now acceptable for publication, you may indicate that here to bypass the “Comments to the Author” section, enter your conflict of interest statement in the “Confidential to Editor” section, and submit your "Accept" recommendation.

Reviewer #1: All comments have been addressed

2. Is the manuscript technically sound, and do the data support the conclusions?

Reviewer #1: Partly

3. Has the statistical analysis been performed appropriately and rigorously? 

Reviewer #1: Yes

4. Have the authors made all data underlying the findings in their manuscript fully available?

Reviewer #1: Yes

5. Is the manuscript presented in an intelligible fashion and written in standard English?

Reviewer #1: No

6. Review Comments to the Author

Reviewer #1: Authors disputed against the reviewers' comments that the number of cases examined is too small. It can be accepted. The reviewer has no other special comments.

7. PLOS authors have the option to publish the peer review history of their article (what does this mean?). If published, this will include your full peer review and any attached files.

Reviewer #1: No

---

## [Editor Report · Acceptance letter]

17 Jan 2020

PONE-D-19-19521R1 

Epidemiology of *Schistosoma mansoni* infection and associated risk factors among School Children attending Primary Schools nearby Rivers in Jimma Town, an urban setting, Southwest Ethiopia 

Dear Dr. Bajiro:

I am pleased to inform you that your manuscript has been deemed suitable for publication in PLOS ONE. Congratulations! Your manuscript is now with our production department. 

With kind regards,

on behalf of

Dr. David Joseph Diemert 

Academic Editor

PLOS ONE